# Radiolabeled Somatostatin Analogs—A Continuously Evolving Class of Radiopharmaceuticals

**DOI:** 10.3390/cancers14051172

**Published:** 2022-02-24

**Authors:** Melpomeni Fani, Rosalba Mansi, Guillaume P. Nicolas, Damian Wild

**Affiliations:** 1Division of Radiopharmaceutical Chemistry, University Hospital Basel, 4031 Basel, Switzerland; rosalba.mansi@usb.ch; 2Division of Nuclear Medicine, University Hospital Basel, 4031 Basel, Switzerland; guillaume.nicolas@usb.ch (G.P.N.); damian.wild@usb.ch (D.W.); 3ENETS Center of Excellence for Neuroendocrine and Endocrine Tumors, University Hospital Basel, 4031 Basel, Switzerland

**Keywords:** somatostatin receptors, neuroendocrine neoplasms, neuroendocrine tumors, agonist, antagonist, somatostatin receptor PET/CT, peptide receptor radionuclide therapy

## Abstract

**Simple Summary:**

Somatostatin receptors (SSTs) are of particular interest in oncology because these proteins are overexpressed on the cell membranes of different human malignancies, especially neuroendocrine tumors (NETs) and neuroendocrine neoplasms (NENs). Radiolabeled short peptide analogs of the natural hormone somatostatin have been developed over the years to target SST-expressing tumors and are used for both imaging (diagnosis) and therapy. Today, this type of radiopharmaceutical plays a pivotal role in the management of NET and NEN patients. Despite their clinical success, new developments in recent years, in terms of peptide analogs and radionuclides, have shown certain advantages and hold promise for further improvement in both the diagnosis and therapy of SST-expressing tumors, even beyond NETs and NENs.

**Abstract:**

Somatostatin receptors (SSTs) are recognized as favorable molecular targets in neuroendocrine tumors (NETs) and neuroendocrine neoplasms (NENs), with subtype 2 (SST_2_) being the predominantly and most frequently expressed. PET/CT imaging with ^68^Ga-labeled SST agonists, e.g., ^68^Ga-DOTA-TOC (SomaKit TOC^®^) or ^68^Ga-DOTA-TATE (NETSPOT^®^), plays an important role in staging and restaging these tumors and can identify patients who qualify and would potentially benefit from peptide receptor radionuclide therapy (PRRT) with the therapeutic counterparts ^177^Lu-DOTA-TOC or ^177^Lu-DOTA-TATE (Lutathera^®^). This is an important feature of SST targeting, as it allows a personalized treatment approach (theranostic approach). Today, new developments hold promise for enhancing diagnostic accuracy and therapeutic efficacy. Among them, the use of SST_2_ antagonists, such as JR11 and LM3, has shown certain advantages in improving image sensitivity and tumor radiation dose, and there is evidence that they may find application in other oncological indications beyond NETs and NENs. In addition, PRRT performed with more cytotoxic α-emitters, such as ^225^Ac, or β^-^ and Auger electrons, such as ^161^Tb, presents higher efficacy. It remains to be seen if any of these new developments will overpower the established radiolabeled SST analogs and PRRT with β^-^-emitters.

## 1. Introduction

The somatostatin family consists of two cyclic disulfide-bond-containing peptide hormones, one with 14 amino acids (SS-14, primary form in the brain) and one with 28 amino acids (SS-28, primary form in the gut). The biologic actions of somatostatin are mediated by five somatostatin receptor subtypes (SST_1-5_), which belong to a distinct group within the G-protein-coupled receptor superfamily, also known as 7-transmembrane receptors. The activation of these receptors stimulates multiple intracellular cascades to modulate growth hormone release, insulin and glucagon secretion, gastric acid secretion, and neuronal activity. The five subtypes (SST_1-5_) have approx. 50% identical amino acids, with homology being the most pronounced in the transmembrane regions, and they are subdivided into two subgroups: one consisting of SST_2_, SST_3_, and SST_5_, differing from the other subgroup, which consists of SST_1_ and SST_4_ in terms of amino acid homology and pharmacological profile [1]. SSTs are of particular interest in oncology, because their expression is linked to different human malignancies [2,3,4,5]. SSTs are recognized as favorable molecular targets in neuroendocrine tumors (NETs) and neuroendocrine neoplasms (NENs) for targeting and drug delivery, with subtype 2 (SST_2_) being the predominantly and most frequently expressed [6,7,8].

Today, radiopharmaceuticals targeting the SST play a pivotal role in the management of NEN and NET patients [6,7]. These radiopharmaceuticals are mainly based on short peptide analogs of the natural hormone somatostatin, and their clinical success lies in the following factors: (a) the expression of SST in a high incidence and density on the surface of NET cells (easily accessible) compared to their low expression in other tissues; (b) the development, over the years, of synthetic peptide analogs of somatostatin, which have been optimized in terms of in vivo stability, affinity, specificity, and pharmacokinetics; and (c) the advances in radiochemistry and chelation chemistry, which have allowed for the chemical tuning of these peptides for radiolabeling with various radionuclides for different medical applications in nuclear oncology. Undoubtedly, radiolabeled somatostatin analogs have paved the way for a number of modern developments, especially for nuclear oncology and endocrinology. This review features the development and application of SST-targeting radiopharmaceuticals, and it represents both the radiochemist’s and the clinician’s view. This article provides a concise overview of the current status, the latest developments, and the future prospects in the field. More precisely, it presents (I) the radiolabeled SST agonists, including the key structural features of somatostatin that led to the currently established radiopharmaceuticals, their clinical applications, and the most recent advancements; (II) the radiolabeled SST antagonists, from their conceptualization and their structural design in comparison with the agonists to the clinical data and status of their development to date; (III) the current evidence for novel clinical indications of radiolabeled SST analogs, especially antagonists; and (IV) the perspectives of labeling with new radionuclides and of targeting somatostatin receptor subtypes other than SST_2_.

## 2. Somatostatin Receptor Agonists: The Archetype and the Latest Developments

### 2.1. Peptide Sequences and Critical Amino Acid Positions

In the amino acid sequence of the endogenous hormone somatostatin, the small tetrapeptide Phe-Trp-Lys-Thr (corresponding to the amino acid residues 7–10 in the natural hormone somatostatin-14) was identified as essential for receptor recognition and biological activity [9,10]. The introduction of d-amino acids for improved in vivo stability and stepwise optimization, based on the minimal amino acid chain length in somatostatin, resulted in an octapeptide with a type II β-turn, formed by the active core Phe-d-Trp-Lys-Thr in a six-member ring via a disulfide bridge, known as octreotide (OC, Table 1) [11]. Octreotide (Sandostatin^®^) is used for the management of growth-hormone-producing tumors (e.g., acromegaly), and tumor and symptom control of neuroendocrine tumors [12], and it has been the starting point for the development of radiolabeled somatostatin analogs (Figure 1). It is worth mentioning that while the natural hormones somatostatin-14 and somatostatin-28 bind to all subtypes with high (though not the same) affinity, short somatostatin analogs, such as octreotide, only bind to the first subgroup of receptor subtypes (Table 1). More precisely, octreotide has high affinity to SST_2_ and SST_5_ and moderate affinity to SST_3_. The most interesting structural features on octreotide-based analogs are position 3 (Phe), which is involved in the critical β-turn, and position 8 (Thr(ol)), modifications of which have led to analogs with different receptor subtype selectivities and affinities. Briefly, the well-known Tyr^3^-octreotide (TOC), where Phe is substituted by Tyr, shows high affinity to SST_2_ and moderate affinity to SST_5_, while 1-Nal^3^-octreotide (NOC) and BzThi^3^-octreotide (BOC) show additional affinity to SST_3_. The analog with substitution in both positions, [Tyr^3^, Thr^8^]-octreotide ([Tyr^3^]-octreotate or TATE), binds almost selectively to SST_2_, while the corresponding [1-Nal^3^, Thr^8^]-octreotide (NOC-ATE) and [BzThi^3^, Thr^8^]-octreotide (BOC-ATE) show additional affinity to SST_5_ and SST_3_ [13,14]. See Table 1 for affinity data.

After the pioneering work of Lamberts et al. in 1989, where endocrine-related tumors could be visualized using ^123^I-labeled Tyr^3^-octreotide (TOC), the conjugation of chelators for labeling with radiometals revolutionized the field (Figure 1) [15]. More specifically, the following advances can be noted: (a) the clinical success of ^111^In-DTPA-octreotide (OctreoScan^®^, where DTPA: diethylenetriaminepentaacetic acid); (b) the introduction of the chelator 1,4,7,10-tetraazacyclododecane-1,4,7,10-tetraacetic acid (DOTA), which is able to form thermodynamically and kinetically stable complexes with a series of 3+ radiometals, like the β^-^-emitter ^90^Y; (c) the introduction of peptide receptor radionuclide therapy (PRRT) with ^90^Y- or ^177^Lu-labeled SST agonists, such as ^90^Y- or ^177^Lu-DOTA-TOC and ^177^Lu-DOTA-TATE [16]; and (d) the accelerated development of ^68^Ga radiochemistry/radiopharmacy, establishing SST PET/CT with somatostatin analogs, such as ^68^Ga-DOTA-TOC, ^68^Ga-DOTA-TATE, and ^68^Ga-DOTA-NOC, allowing the most sensitive staging and restaging of NETs, as well as the identification of patients who would benefit from PRRT (theranostic approach), which made radiolabeled somatostatin analogs the archetype of peptide-based radiopharmaceuticals. Nowadays, a plethora of radiolabeled somatostatin analogs have been developed in order to optimize affinity, specificity, and/or pharmacokinetics (many reviews are available, see, for example, Eychenne R et al. [17]). Among them, DOTA-TOC and DOTA-TATE remain the most widely used analogs, with DOTA-TATE (NETSPOT^®^) and DOTA-TOC (SomaKit TOC^®^) kits having approval by the US Food and Drug Administration (FDA) and European Medicines Agency (EMA) for ^68^Ga-labeling, and ^177^Lu-DOTA-TATE (^177^Lu-oxodotreotide or Lutathera^®^) being the only agent approved for therapy to date. It is expected that the approval of ^177^Lu-DOTA-TOC (^177^Lu-edotreotide) will follow the completion of the COMPETE (NCT03049189) phase III trial.

### 2.2. Clinical Studies and Approvals

Today, PRRT with radiolabeled SST agonists (e.g., DOTA-TOC or DOTA-TATE, Table 1) is part of the standard of care of NENs. NETTER-1 (NCT01578239; EudraCT number 2011-005049-11) was the first prospective, open-label, randomized, phase III trial to compare four cycles of ^177^Lu-DOTA-TATE (4 × 7.4 GBq) plus 30 mg long-acting release octreotide (PRRT group, n = 117) with high-dose (60 mg double dose) long-acting release octreotide (control group, n = 114) in advanced, progressive midgut NET patients. There was a significantly longer progression-free survival for the PRRT arm (*p* < 0.001) [22] and a significant improvement in quality of life [23]. Consequently, ^177^Lu-DOTA-TATE (^177^Lu-oxodotreotide) received marketing authorization for the treatment of adult patients with SST-positive gastroenteropancreatic neuroendocrine tumors (GEP-NETs). At the final analysis of overall survival (OS), the median OS was improved by 11.7 months for the ^177^Lu-DOTA-TATE arm versus the control arm (48.0 (95% CI, 37.4–55.2) vs. 36.3 (95% CI, 25.9–51.7) months, respectively), which, however, did not reach statistical significance in the long-term follow-up with a median of 6.3 years [24]. Regarding safety, the NETTER-1 data show a low incidence of long-term side-effects regarding hematotoxicity and nephrotoxicity.

Currently, a second prospective, randomized, controlled, open-label, multi-center, phase III trial, COMPETE (NCT03049189), is ongoing, in which PRRT using ^177^Lu-DOTA-TOC (^177^Lu-edotreotide, four cycles with 7.5 GBq/cycle) is being compared with the mTOR inhibitor everolimus (10 mg daily) in patients with progressive, SST-positive GEP-NETs. Upon completion of the study, the approval of ^177^Lu-DOTA-TOC is expected. These trials and other trials (e.g., OCCLURANDOM, NCT02230176) should further precisely determine the position of PRRT in the current clinical algorithm with regard to other systemic therapies, such as everolimus and sunitinib.

Using routes other than intravenous administration may be an interesting approach to enhance the therapeutic and safety window of PRRT. NENs and their liver metastases are often highly perfused, and the intra-arterial route can exploit the first-pass effect to treat liver-dominant disease more efficiently. Such an approach can also be used for inoperable primary tumors to downstage the disease in the neoadjuvant setting [25,26]. However, large comparative prospective trials supporting its wider use are missing.

### 2.3. Combination with Alpha-Emitters

Alpha particles have a very short range in tissues (20–100 μm), irradiating volumes with cellular dimensions and therefore sparing normal surrounding tissues from cytotoxic radiation. At the same time, their linear energy transfer (LET) is much higher compared to that of β^-^ particles (50–230 vs. 0.2 keV/μm), which makes alpha radiation far more cytotoxic. Among the α-emitters, ^213^Bi was initially used in combination with DOTA-TOC. Kratochwil et al. performed the first clinical study (retrospective) with an α-emitter in combination with DOTA-TOC (^213^Bi-DOTA-TOC) in seven patients with metastatic NETs (activities ranging from 3.3 to 21 GBq in one–five cycles) after progressing under ^90^Y-/^177^Lu-DOTA-TOC therapy, and it is already available [27]. The report showed moderate renal and hematological toxicity but possible long-term bone marrow toxicity, with the diagnosis of multiple dysplastic syndrome/acute myeloid lymphoma MDS/AML in one heavily pretreated patient. However, the current supply limitations of high-activity ^225^Ac/^213^Bi generators have prevented larger confirmatory prospective studies and have instead motivated the use of the α-emitters ^225^Ac or ^212^Pb.

^212^Pb-DOTAMTATE (AlphaMedix™) is in a phase I, non-randomized, open-label, dose-escalation, single-center study in 20 PRRT naïve NET patients (NCT03466216), with the highest dose level being four cycles of 2.50 MBq/kg/cycle. Previously, at the highest dose level in a small cohort of 10 NET patients, the objective radiological response (ORR) was 80%, and it had mild adverse effects and a tolerable safety profile [28].

^225^Ac-DOTA-TOC was administered in 40 patients with progressive NENs, where the maximum tolerated dose was established at 40 MBq as a single fraction and at 25 MBq in two fractions at a 4-month interval [29]. In another study, ^225^Ac-DOTA-TATE was reported in 32 patients with metastatic GEP-NETs, who were stable or had progressive disease and were on ^177^Lu-DOTA-TATE therapy. After the administration of 7.8–44.4 MBq ^225^Ac-DOTA-TATE in one–five portions, partial remission was achieved in 15 patients and stable disease in 9 of them. At a median of 8-month follow-up, no disease progression or deaths were documented [30]. Recently, a retrospective analysis was performed in 39 patients who received ^225^Ac-DOTA-TOC in an attempt to define the safety levels of ^225^Ac-DOTA-TOC [31]. The analysis was mainly conclusive regarding acute hematological toxicity but not regarding chronic nephrotoxicity due to pre-existing risk factors. Overall, it was found that a single dose of up to 29 MBq, repeated doses of ~20 MBq in 4-month intervals, and a cumulative dose of 60–80 MBq were hematologically tolerable and avoided high-grade (3/4) hematotoxicity.

Although α-emitters offer potential advantages over β^-^-emitters therapeutically, long-term toxicity data are still lacking to properly assess the therapeutic benefit. Importantly, the translocation of radioactive daughter nuclides from the chelator should also be considered as a potential safety hazard for α-emitters with multiple α-emitting daughters, such as ^225^Ac.

### 2.4. Conjugates with Prolonged Circulation

Despite the successful outcome of the NETTER-1 study, the objective response rate of patients treated with ^177^Lu-DOTA-TATE was, at most, 18% [22], probably due to the rapid blood clearance of ^177^Lu-DOTA-TATE, leading to a suboptimal tumor residence time. An attempt to overcome this drawback was the incorporation of Evans Blue (EB) motifs, which prolongs the half-life of the conjugate in the blood by having low micromolar affinity to albumin. This concept was applied to DOTA-TATE, for which it was shown that treatment with ^177^Lu-DOTA-EB-TATE was more effective in SST_2_-expressing xenografts than ^177^Lu-DOTA-TATE [32,33]. The first dosimetry data of the long-circulating SST_2_ agonist ^177^Lu-DOTA-EB-TATE versus ^177^Lu-DOTA-TATE showed a 7.9-fold increase in tumor dose, which was counterbalanced with an even greater increase in renal and bone marrow absorbed doses [34]. A better response rate (assessed by ^68^Ga-DOTA-TATE PET/CT) after one cycle of treatment was reported for the EB conjugate [35], but matching doses in the kidneys and bone marrow were not provided. The superiority of ^177^Lu-DOTA-EB-TATE was not confirmed in an intraindividual comparison versus ^177^Lu-DOTA-TOC in a limited number (n = 5) of patients [36], where the tumor-to-critical organs’ absorbed dose ratios (defined as therapeutic index) were mainly higher for ^177^Lu-DOTA-TOC and not for ^177^Lu-DOTA-EB-TATE. Whether ^177^Lu-DOTA-EB-TATE has any benefit over the established radiopharmaceuticals is still debatable.

## 3. Somatostatin Receptor Antagonists: Will They Make the Difference?

### 3.1. Preclinical Development

The observation that GPCR antagonists may bind to more binding sites than agonists, since their binding is independent of the fraction of receptors coupled to the GTP-binding proteins [37], was the primary reason for the development of radiolabeled SST antagonists. Structurally, the main feature to convert an agonist to an antagonist was shown to be the inversion of chirality at positions 1 and 2 of the octreotide family [38]. From the very first preclinical evaluation, the superiority of radiolabeled SST antagonists over agonists was illustrated in terms of targeting SST-expressing tumors [20]. For example, the first SST_2_ antagonist ^111^In-DOTA-BASS (Table 1) showed almost twice higher tumor uptake compared to the agonist ^111^In-DTPA-TATE, despite its lower affinity (IC_50_ = 9.4 ± 0.4 nM vs. 1.3 ± 0.2 nM [18,20]), and it also showed binding to a higher number of sites on the cell membrane (B_max_) [20]. This was, further on, confirmed on human tumor tissues by autoradiography when comparing ^177^Lu-DOTA-BASS with ^177^Lu-DOTA-TATE [39], and the long-lasting tumor uptake of ^177^Lu-DOTA-BASS in xenografts in vivo holds promise for therapeutic applications of the antagonists [40]. A series of analogs were developed by systematic substitutions of different amino acids with the aim of identifying the structural features that lead to SST_2_-selective antagonists with high affinity [41]. The analogs JR11 and LM3 (Table 1) were selected among the ones with the best affinity and highest hydrophilicity, and they were studied in combination with different chelators and various radiometals [21,42].

Several reports in the past had shown that adding a radiometal to a chelator–SST agonist conjugate could alter its affinity, with ^68^Ga systematically improving the SST_2_ affinity of DOTA-conjugated agonists, as well as their pharmacokinetics, compared to ^111^In, ^90^Y, and ^177^Lu [13,18]. The effect of the radiometal, but also of the chelator, was far more impressive, and even unexpected, for the SST_2_ antagonists [21,42]. Comprehensive studies with JR11 and LM3 in combination with different chelators, such as DOTA and NODAGA, and various (radio)metals, including Ga, Cu, In, Y, and Lu, have illustrated a very high sensitivity of the SST_2_ antagonists to the N-terminal modification needed for radiolabeling, and they have shed light on the most promising metal–chelator–antagonist combinations for further development, having the following major impacts: (1) All Ga-DOTA conjugates lost affinity for SST_2_, contrary to the (radio)metalated In-, Y-, and Lu-DOTA conjugates. The affinity of the Ga-complexes was recovered by replacing DOTA with NODAGA. For instance, ^68^Ga-NODAGA-LM3 has a 10-fold higher SST_2_ affinity than ^68^Ga-DOTA-LM3, and ^68^Ga-NODAGA-JR11 has an almost 25-fold higher affinity than ^68^Ga-DOTA-JR11 (Table 1). Therefore, ^68^Ga-NODAGA conjugates of SST_2_ antagonists were selected for clinical development. (2) The great potential of using SST_2_ antagonists became obvious when the low-affinity ^68^Ga-DOTA-JR11 was compared to ^68^Ga-DOTA-TATE, which had approx. a 150-fold higher affinity (Table 1). It was found in vivo that ^68^Ga-DOTA-JR11 outweighed the affinity differences, being even slightly better than the high-affinity ^68^Ga-DOTA-TATE. Not to mention that the high affinity ^68^Ga-NODAGA-JR11 was better distinguished than ^68^Ga-DOTA-TATE in terms of tumor uptake [21].

Similarly, the therapeutic counterpart ^177^Lu-DOTA-JR11 compared to ^177^Lu-DOTA-TATE showed a higher tumor uptake and, more importantly, a longer tumor residence time, leading to a higher radiation tumor dose [43] and, consequently, delayed tumor growth and longer median survival [44]. The reasons for these observed in vivo differences can be found, at least partially, in the differences between the two radiopharmaceuticals on the cellular level, which were recently investigated [45]. ^177^Lu-DOTA-JR11 showed faster association, slower dissociation, and longer cellular retention than ^177^Lu-DOTA-TATE. Despite a comparable high affinity, ^177^Lu-DOTA-JR11 recognized four times more receptor binding sites than ^177^Lu-DOTA-TATE. However, more interestingly, while a high excess of antagonist was able to entirely displace the agonist bound on the cell membrane, the agonist could not completely displace the antagonist. Taken together, the antagonist binds not only to additional binding sites but also to different binding sites that are not recognized by the agonist (e.g., uncoupled G proteins) [45]. This observation is clinically relevant, as it indicates that the interruption of somatostatin agonists before treatment with radiolabeled analogs may not be necessary if SST_2_ antagonists are used.

Last but not least, SPECT tracers based on antagonists are missing, but they are also important considering that more than 70% of nuclear medicine procedures still use ^99m^Tc. The first attempts to label SST_2_ antagonists with ^99m^Tc via the monodentate ligand hydrazinonicotinamide (HYNIC) using ethylenediamine *N*,*N*′ diacetic acid (EDDA) as a co-ligand (similarly to the clinically used agonist [^99m^Tc]Tc-HYNIC/EDDA-TOC) failed because the antagonist entirely lost its affinity for SST_2_ [46], once more depicting the extreme sensitivity of the antagonists to N-terminal modifications. Further studies illustrated that the loss of affinity can be circumvented, to a certain extent, when a spacer of appropriate length and nature (e.g., aminohexanoic acid) is introduced between the antagonist and HYNIC [47]. Nevertheless, the alternative chelating system 6-carboxy-1,4,8,11-tetraazaundecane (N4) seems to be better suited to ^99m^Tc-based SST_2_ antagonists. In fact, ^99m^Tc-labeled LM3 via N4 ([^99m^Tc]Tc-TECANT-1) has been selected as the first ^99m^Tc-based antagonist for clinical translation [48] under the ERAPerMED project “TECANT” (Ref No. ERAPERMED2018-125). The clinical trial is expected to start soon.

### 3.2. Clinical Translation

The first clinical evidence indicating that imaging with SST_2_ antagonists may be superior to that with agonists was provided by a prospective study, which included five patients with NETs or thyroid cancer after total-body scintigraphy and a SPECT/CT scan with ^111^In-DOTA-BASS versus OctreoScan [49]. ^111^In-DOTA-BASS had a higher tumor detection rate (25/28 lesions) than ^111^In-DTPA-octreotide (17/28 lesions) in a lesion-based analysis. Meanwhile, based on affinity studies and preclinical results, ^68^Ga-NODAGA-JR11 (=^68^Ga-OPS202) was selected for PET/CT imaging studies. Nicolas et al. performed a single-center, prospective, phase I/II study with 12 GEP-NET patients, comparing PET/CT with two micro doses of ^68^Ga-NODAGA-JR11 (15 and 50 μg/150 MBq) and one micro dose of the potent SST_2_ agonist ^68^Ga-DOTA-TOC (NCT02162446). ^68^Ga-NODAGA-JR11 showed favorable dosimetry results and imaging properties, with the best tumor contrast between 1 and 2 h after injection [50]. ^68^Ga-NODAGA-JR11 PET/CT showed a significantly higher sensitivity in a lesion-based comparison with ^68^Ga-DOTA-TOC PET/CT: 93.7% (95% CI: 85.3–97.6%) vs. 59.2% (95% CI: 36.3–79.1%) [51]. In this study, diagnostic efficacy measures were compared against contrast-enhanced CT or MRI. ^68^Ga-DOTA-JR11 was also assessed clinically, despite its >20 times lower affinity compared to ^68^Ga-NODAGA-JR11 (Table 1) [21,52,53,54]. Zhu et al. prospectively compared ^68^Ga-DOTA-JR11 and ^68^Ga-DOTA-TATE PET/CT in the same patients with NETs [54]. As in the study of Nicolas et al., they detected significantly more liver lesions with the SST_2_ antagonist (552 vs. 365) but, at the same time, significantly less bone lesions (158 vs. 388) compared to ^68^Ga-DOTA-TATE. Importantly, ^68^Ga-DOTA-JR11 showed a lower tumor uptake than ^68^Ga-DOTA-TATE, which is in contrast to the study of Nicolas et al., who prospectively compared ^68^Ga-NODAGA-JR11 and ^68^Ga-DOTA-TOC PET/CT in the same patients [51]. This finding can be explained by the much lower SST_2_ affinity of ^68^Ga-DOTA-JR11 in comparison to ^68^Ga-NODAGA-JR11 (Table 1) and/or by the study design, which may have caused a bias, as ^68^Ga-DOTA-TATE PET/CT was always performed 24 h ahead of ^68^Ga-DOTA-JR11 PET/CT, creating the risk of receptor occupation and/or internalization [55].

The therapeutic companion ^177^Lu-DOTA-JR11 (=^177^Lu-OPS201), which was initially assessed in a single-center, prospective, proof-of-principle study (phase 0 study), was compared with ^177^Lu-DOTA-TATE in the same four patients with advanced, metastatic neuroendocrine neoplasia (NEN) (grades 1–3) [56]. The median tumor dose was 3.5-fold higher for the antagonist. At the same time, tumor-to-kidney dose ratios were >2-fold higher with ^177^Lu-DOTA-JR11 compared to ^177^Lu-DOTA-TATE. Overall, tumor doses with ^177^Lu-DOTA-JR11 were up to 487 Gy, with moderate adverse events with grade 3 thrombocytopenia after treatment with three cycles (total 15.2 GBq) in one patient. Figure 2 illustrates a direct comparison of the antagonist ^177^Lu-DOTA-JR11 versus the agonist ^177^Lu-DOTA-TOC in the same patient with lung NETs (G2).

Later on, a single-center phase I study with 20 NET patients (grades 1–3) reported a best overall response (RECIST 1.1 criteria) of 45%, and the median progression-free survival (PFS) was 21 months (95% CI, 13.6-NR), accompanied, however, with grade 4 hematotoxicity (leukopenia, neutropenia, and thrombocytopenia) in four out of seven patients treated with two cycles of ^177^Lu-DOTA-JR11 (cumulative activity between 10.5 and 14.7 GBq) [57]. Hence, the study was suspended, and the protocol was modified to limit the cumulative absorbed bone marrow dose. ^177^Lu-DOTA-JR11 (^177^Lu-OPS201) is currently being evaluated in a phase I/II, multi-center, open-label study (NCT02592707—active, not recruiting). To date, there is only an abstract available with a brief summary of the results of 20 NET patients with an adequate follow-up [58]. The disease control rate (DCR) at 12 months was 90% (95% CI: 68.3–98.8) for these 20 patients.

More recently, in parallel to the development of the theranostic pair based on JR11, the other antagonist, LM3, was also developed. Results in the form of abstracts have reported the feasibility of PET/CT imaging with ^68^Ga-NODAGA-LM3 in 40 patients with GEP-NET, lung NET, paraganglioma/pheochromocytoma, etc. [59], and a higher detection rate of ^68^Ga-NODAGA-LM3 versus ^68^Ga-DOTA-TOC PET/CT in 10 paraganglioma patients, with ^68^Ga-NODAGA-LM3 PET/CT detecting many more lesions (243 vs. 177), including bone lesions (190 vs. 143) [60]. Meanwhile, ^68^Ga-NODAGA-LM3 and ^68^Ga-DOTA-LM3 were compared in a randomized, double-blind study with 16 NET patients [61]. The SUVmax values of tumors and SST_2_-positive organs were >2 times higher with ^68^Ga-NODAGA-LM3 than with ^68^Ga-DOTA-LM3 at 2 h post-injection, which is consistent with the almost 10 times higher SST_2_ affinity of ^68^Ga-NODAGA-LM3 compared to ^68^Ga-DOTA-LM3 (Table 1) [21].

The therapeutic companion, ^177^Lu-DOTA-LM3, was evaluated in a single-center, compassionate-use study, which included 51 patients with metastatic NENs of grades 1–3, who were selected after ^68^Ga-NODAGA-LM3 PET/CT imaging [62]. There were few adverse events (maximal grade 3 thrombocytopenia in 5.9% of patients) after treatment with one–four cycles of ^177^Lu-DOTA-LM3, with mean cumulative activity between 6.1 and 26.1 GBq. The partial response and DCR (RECIST 1.1 criteria in 47 patients) were 36% and 85% at 3–6 months, respectively.

## 4. Novel Indications for Radiolabeled Somatostatin Analogs

The binding capacities of radiolabeled SST antagonists and agonists were compared in human tissue samples from nine different tumors using in vitro autoradiography with ^177^Lu-DOTA-BASS vs. ^177^Lu-DOTA-TATE [39], as mentioned above, and with ^125^I-JR11 vs. ^125^I-TOC [63]. A summary of the outcome is provided in Figure 3.

In all tested cases, the radiolabeled SST_2_ antagonist bound to more SST_2_ sites in all tumors, with an uptake that was 3.8–21.8 times higher than that with the agonist. Interestingly, in some non-neuroendocrine neoplasias, the level of binding of the antagonists reached the same level as that of the agonists (e.g., ^177^Lu-DOTA-TATE) in well-differentiated NENs. Of particular interest is the fact that tumors other than GEP-NETs and lung NETs have the potential to become targets for radiolabeled SST_2_ antagonists, despite the relatively low SST_2_ expression, for example, non-Hodgkin lymphomas, renal cell carcinoma, breast cancer, pheochromocytoma, paraganglioma, medullary thyroid cancer, small-cell lung cancer, and paraganglioma.

SSTs are also expressed in peritumoral vessel endothelial cells; in inflammatory cells; and in immune system cells, such as activated lymphocytes, monocytes, and epithelioid cells. This suggests that clinical indications can be found in benign and chronic inflammatory diseases, besides oncology [64]. PET imaging with ^68^Ga-labeled SST agonists (DOTA-TOC, DOTA-TATE, and DOTA-NOC) have shown relevance in detecting vulnerable, atherosclerotic plaques and have been correlated to other risk factors in patients (summarized in [65]). The use of antagonists in this context has, to date, only been explored preclinically [66]. In terms of PRRT, a retrospective analysis of a limited number of oncological patients indicated that ^177^Lu-DOTA-TATE results in a reduction in atherosclerotic plaque activity [67], while ^177^Lu-DOTA-TOC showed treatment effects in a feasibility study involving two patients with refractory multi-organ involvement of sarcoidosis [68]. Nevertheless, radiolabeled somatostatin analogs have not yet found clinical relevance in these indications, and their impact on clinical outcome needs to be assessed in large-scale clinical trials.

## 5. Perspectives

The use of alternative theranostic pairs of radionuclides, such as radioisotopes of scandium (^43/44/47^Sc) and terbium (^149/152/155/161^Tb), might open novel theranostic applications. Recently, a preclinical study demonstrated clear therapeutic benefit when using ^161^Tb instead of ^177^Lu in combination with SST analogs; ^161^Tb has similar decay properties to ^177^Lu but, additionally, emits a substantial number of conversion and Auger electrons [69]. The most important finding of the study was the identification of the cellular localization of the ^161^Tb-labeled SST analog, which leads to the best therapeutic outcome. It was shown that the combination of ^161^Tb with the SST_2_ antagonist DOTA-LM3, which is not internalizing but remains on the cell membrane, was a better combination than the internalized cytoplasm agonist DOTA-TOC and the internalized and partially nucleus-localized DOTA-TOC-NLS bearing a nucleus-targeting unit (nuclear localization signal (NLS)) [69]. Overall, the preclinical data suggest a benefit of treating NENs with ^161^Tb-DOTA-LM3 (or ^161^Tb-labeled SST antagonists) vs. ^177^Lu-DOTA-TOC (^161^Tb-labeled SST agonists).

To date, radiolabeled somatostatin analogs used for treatment bind with high affinity to the most predominantly expressed SST_2_. However, various expression and co-expression patterns have been described for the five somatostatin receptor subtypes (SST_1-5_), depending on the tumor type and origin [3,70,71]. Interestingly, tumor areas lacking expression of a given subtype may be populated by another one [70,71]. In addition, the downregulation or loss of SST_2_ in advanced disease stages is associated with an inherently worse disease prognosis, a lower sensitivity in imaging, and ineffective therapy with SST_2_-specific analogs due to inadequate tumor targeting. Hence, somatostatin analogs with affinity to more than one receptor subtype are of great interest, as they address receptor subtype co-expression and heterogeneous expression patterns [72]. Analogs targeting more subtypes than SST_2_ potentially target a broader spectrum of tumors and/or increase the uptake of a given tumor and are, therefore, a field to explore.

## Figures and Tables

**Figure 1 cancers-14-01172-f001:**
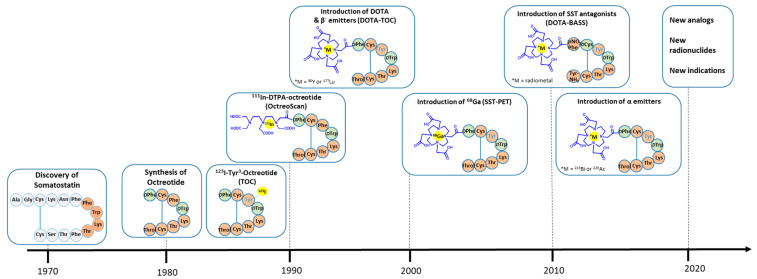
Evolution in the development of radiolabeled somatostatin analogs. Color code: orange for L-amino acids (also showing the essential amino acids (tetrapeptide) in the somatostatin sequence for receptor recognition), green for D-amino acids, blue for chelators, and yellow for radionuclides.

**Figure 2 cancers-14-01172-f002:**
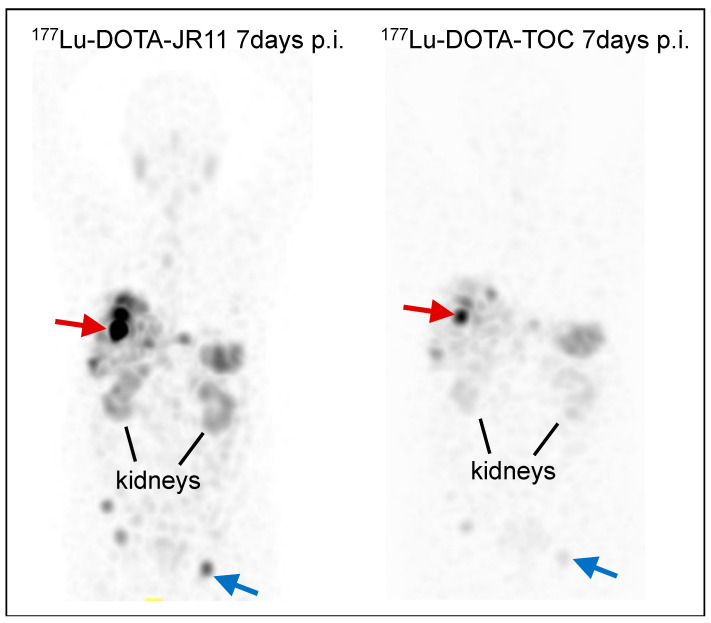
Multi-intensity projection SPECT of ^177^Lu-DOTA-JR11 and ^177^Lu-DOTA-TOC in the same patient with metastatic atypical lung carcinoid. Arrows show tumor doses of ^177^Lu-DOTA-JR11 vs. ^177^Lu-DOTA-TOC: red arrow: 12.6 vs. 3.36 Gy/GBq, blue arrow: 9.89 vs. 1.46 Gy/GBq.

**Figure 3 cancers-14-01172-f003:**
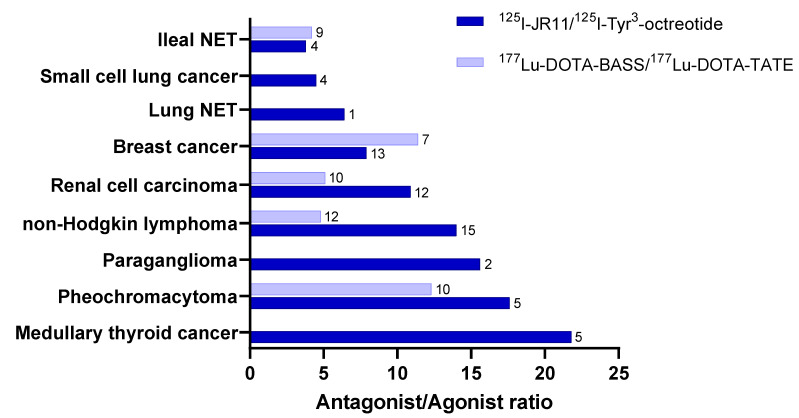
Radiolabeled SST_2_ agonist/antagonist binding in different human tumors. ^125^I-JR11/^125^I-Tyr^3^-octreotide data are from [63]. ^177^Lu-DOTA-BASS/^177^Lu-DOTA-TATE data are from [39]. Numbers indicate the samples sizes.

**Table 1 cancers-14-01172-t001:** Somatostatin-based radiotracers. Affinity data (IC_50_ = half maximal inhibitory concentration) and clinical status.

Amino Acid Sequence	Radiotracer	IC_50_ (nM ± SEM)			Clinical Status
		SST_2_	SST_3_	SST_5_	
Agonists					
D-Phe-c(Cys-Phe-D-Trp-Lys-Thr-Cys)Thr(ol)	^111^In-DTPA-OC *	22 ± 3.6	182 ± 13	237 ± 52	FDA/EMA approved
D-Phe-c(Cys-Tyr-D-Trp-Lys-Thr-Cys)Thr(ol)	^68^Ga-DOTA-TOC *	2.5 ± 0.5	613 ± 140	73 ± 21	Prospective phase II
	^90^Y-DOTA-TOC *	11 ± 1.7	389 ± 135	114 ± 29	Clinical data
	^177^Lu-DOTA-TOC	n.r.	n.r.	n.r.	Prospective phase III
D-Phe-c(Cys-Tyr-D-Trp-Lys-Thr-Cys)Thr	^68^Ga-DOTA-TATE *	0.2 ± 0.04	>1′000	377 ± 18	FDA/EMA approved
	^177^Lu-DOTA-TATE ^#^	2.0 ± 0.8	162 ± 16	>1′000	FDA/EMA approved
D-Phe-c(Cys-1-Nal-D-Trp-Lys-Thr-Cys)Thr(ol)	^68^Ga-DOTA-NOC ^&^	1.9 ± 0.4	40 ± 5.8	7.2 ± 1.6	Prospective phase II
Antagonists					
p-NO_2_-Phe-c(D-Cys-Tyr-D-Trp-Lys-Thr-Cys)Tyr-NH_2_	^111^In-DOTA-BASS ^¥^	9.4 ± 0.4	>1000	>1000	Preliminary clinical data
p-Cl-Phe-c(D-Cys-Tyr-D-Aph(Cbm)-Lys-Thr-Cys)Tyr-NH_2_	^68^Ga-DOTA-LM3 ^¶^	12.5 ± 4.3	>1000	>1000	Prospective phase I/II
	^68^Ga-NODAGA-LM3 ^¶^	1.3 ± 0.3	>1000	>1000	Prospective phase I/II
	^177^Lu-DOTA-LM3	n.r.	n.r.	n.r.	Preliminary clinical data
p-Cl-Phe-c(D-Cys-Aph(Hor)-D-Aph(Cbm)-Lys-Thr-Cys)Tyr-NH_2_	^68^Ga-NODAGA-JR11 ^¶^	1.2 ± 0.2	>1000	>1000	Prospective phase I/II
	^68^Ga-DOTA-JR11 ^¶^	29 ± 2.7	>1000	>1000	Prospective theranostic
	^177^Lu-DOTA-JR11 ^¶^	0.7 ± 0.2	>1000	>1000	Prospective phase I/II

1-Nal = 1-naphthyl-alanine; Aph(Hor) = 4-amino-L-hydroorotyl-phenylalanine; D-Aph(Cbm) = D-4-amino-carbamoyl-phenylalanine. n.r. = not reported. * Data are from [18]; ^#^ Data are from [19] (different lab); ^&^ Data are from [13]; ^¥^ Data are from [20]; ^¶^ Data are from [21].

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
