# Peer review of "Radiolabeled Somatostatin Analogs—A Continuously Evolving Class of Radiopharmaceuticals"

_cancers, 2022, doi:10.3390/cancers14051172_

Round 1
Reviewer 1 Report
A very interesting and well-prepared review publication. The authors presented in detail somatostatin analogues showing affinity for sst receptors, labeled with both diagnostic (68Ga, 99mTc) and therapeutic (90Y, 177Lu, 225Ac, 213Bi) receptors. Interesting is comparison of the properties of peptide antagonists and agonists of sst receptors. The work will be of use to a wide range of readers. It can be published in present form.
Author Response
Thank you for appreciating our article.
Reviewer 2 Report
The review titled "Radiolabeled Somatostatin Analogs – a Continuing Evolving Class of Radiopharmaceuticals" focused on radiolabeled Somatostatin analogs as theranostic application for somatostatin receptors expressing tumors. The authors provided decent background to the topic. In this review, the authors nicely summarized the evolution of Somatostatin to its clinical application. Overall, this is a well organized and well written review on the selected topic.
Minor suggestion, it is hard to read the figure 1 tables, please consider changing font size or color.
Author Response
Thank you for the suggestion. Font size was increased and the color codes is now clarified in the figure legend.
Reviewer 3 Report
The authors report an overview on the development of radiolabeled somatostatin analogs, clinical translation and future prospective.
The document is well organized with an objective description on the design of somatostatin agonists and antagonists, from the preclinical studies to their clinical translation for diagnostic applications.
The authors also address concisely other applications of the radiolabeled peptides for radiotherapy in the clinical setting, spamming from cancers with high receptor expression to cancers with heterogenous and low receptor expression. The manuscript also highlights strategies to improve pharmacokinetics of the radiotracers for clinical applications.
Overall, the authors cover all the advances in the field of radiolabeled somatostatin analogs in the clinical setting.
Minor comments:
The authors should highlight the main contribution of this document to others in the field.
The review is easy to read, however, the authors should provide a brief description of the review content in the introduction.
Author Response
Thank you for the suggestion. The last paragraph of the introduction was adjusted accordingly (see page 2, lines 68-78).